# Selected Transesophageal Echocardiographic Parameters of Left Ventricular Diastolic Function Predict Length of Stay Following Coronary Artery Bypass Graft—A Prospective Observational Study

**DOI:** 10.3390/jcm11143980

**Published:** 2022-07-08

**Authors:** Samhati Mondal, Nauder Faraday, Wei Dong Gao, Sarabdeep Singh, Sachidanand Hebbar, Kimberly N. Hollander, Thomas S. Metkus, Lee A. Goeddel, Maria Bauer, Brian Bush, Brian Cho, Stephanie Cha, Stephanie O. Ibekwe, Domagoj Mladinov, Noah S. Rolleri, Laeben Lester, Jochen Steppan, Rosanne Sheinberg, Nadia B. Hensley, Anubhav Kapoor, Jeffrey M. Dodd-o

**Affiliations:** 1Department of Anesthesiology, University of Maryland School of Medicine, Baltimore, MD 21201, USA; kimberlyhollander@som.umaryland.edu; 2Department of Anesthesiology and Critical Care Medicine, Johns Hopkins University, Baltimore, MD 21287, USA; nfarada1@jhmi.edu (N.F.); wgao3@jhmi.edu (W.D.G.); shebbar3@jhmi.edu (S.H.); lgoedde@jhu.edu (L.A.G.); mbauer14@jhmi.edu (M.B.); bbush9@jhmi.edu (B.B.); bcho2@jhmi.edu (B.C.); scha4@jhmi.edu (S.C.); llester4@jhmi.edu (L.L.); jsteppan@jhmi.edu (J.S.); rsheinberg@jhmi.edu (R.S.); nhensle2@jhmi.edu (N.B.H.); jdoddo@jhmi.edu (J.M.D.-o.); 3Johns Hopkins Medicine, Baltimore, MD 21287, USA; ssingh77@jhmi.edu; 4Division of Cardiology, Department of Medicine, Johns Hopkins University, Baltimore, MD 21287, USA; tmetkus1@jhmi.edu; 5Department of Anesthesiology, Cardiovascular Division, BTGH, Baylor College of Medicine, Houston, TX 77030, USA; stephanieopusunju@gmail.com; 6Department of Anesthesiology and Critical Care Medicine, University of Alabama, Birmingham, AL 35233, USA; dmladinov@gmail.com; 7Department of Anesthesiology and Perioperative Medicine, University of Pittsburgh Medical Center, Pittsburgh, PA 15213, USA; nsr5002@gmail.com; 8Department of Anesthesiology, Mercy General Hospital, Sacramento, CA 95819, USA; anubhavkapoor@gmail.com

**Keywords:** echocardiographic, nonsystolic, diastolic dysfunction, transesophageal echocardiography, heart failure, coronary artery bypass, perioperative

## Abstract

(1) Importance: Abnormal left ventricular (LV) diastolic function, with or without a diagnosis of heart failure, is a common finding that can be easily diagnosed by intra-operative transesophageal echocardiography (TEE). The association of diastolic function with duration of hospital stay after coronary artery bypass (CAB) is unknown. (2) Objective: To determine if selected TEE parameters of diastolic dysfunction are associated with length of hospital stay after coronary artery bypass surgery (CAB). (3) Design: Prospective observational study. (4) Setting: A single tertiary academic medical center. (5) Participants: Patients with normal systolic function undergoing isolated CAB from September 2017 through June 2018. (6) Exposures: LV function during diastole, as assessed by intra-operative TEE prior to coronary revascularization. (7) Main Outcomes and Measures: The primary outcome was duration of postoperative hospital stay. Secondary intermediate outcomes included common postoperative cardiac, respiratory, and renal complications. (8) Results: The study included 176 participants (mean age 65.2 ± 9.2 years, 73% male); 105 (60.2%) had LV diastolic dysfunction based on selected TEE parameters. Median time to hospital discharge was significantly longer for subjects with selected parameters of diastolic dysfunction (9.1/IQR 6.6–13.5 days) than those with normal LV diastolic function (6.5/IAR 5.3–9.7 days) (*p* < 0.001). The probability of hospital discharge was 34% lower (HR 0.66/95% CI 0.47–0.93) for subjects with diastolic dysfunction based on selected TEE parameters, independent of potential confounders, including a baseline diagnosis of heart failure. There was a dose–response relation between severity of diastolic dysfunction and probability of discharge. LV diastolic dysfunction based on those selected TEE parameters was also associated with postoperative cardio-respiratory complications; however, these complications did not fully account for the relation between LV diastolic dysfunction and prolonged length of hospital stay. (9) Conclusions and Relevance: In patients with normal systolic function undergoing CAB, diastolic dysfunction based on selected TEE parameters is associated with prolonged duration of postoperative hospital stay. This association cannot be explained by baseline comorbidities or common post-operative complications. The diagnosis of diastolic dysfunction can be made by TEE.

## 1. Introduction

Risk scoring systems for cardiac surgery patients first became popular with the Parsonnet score to calculate mortality risk [1]. Though subsequent evolution allows for calculation of postoperative morbidity [2] and reflects advances in our understanding of the patient’s determinants for risk [3], these risk scoring systems remain imperfect [4,5].

Heart failure (HF) is increasingly becoming a worldwide health burden, and the USA is no exception [6,7,8]. A significant number of heart failure patients have preserved ejection fraction (EF), but abnormal diastolic function [9]. Hence, HF can be broadly subdivided into two categories—systolic (reduced EF/HFrEF) and diastolic or with preserved EF (HFpEF). It is now recognized that cardiac performance during diastole influences morbidity [10,11,12]. The most recent joint guidelines on HF by American Heart Association/American College of Cardiology/Heart Failure Society of America (AHA/ACC/HFSA) have stressed upon abnormal diastolic function on imaging as an indicator of increased LV filling pressure in case of HFpEF [8]. However clear expert guidelines stratifying perioperative risks of diastolic dysfunction, particularly those undergoing cardiac surgery is still lacking [10,11].

Coronary artery disease and myocardial ischemia can affect diastolic function and often precedes systolic dysfunction. That is why many patients who have severe CAD requiring coronary artery bypass (CAB) may have diastolic dysfunction yet preserved EF [12]. Kim et al. conducted an observational study to evaluate implication of diastolic function in long term outcomes of patients with CAD after percutaneous coronary intervention (PCI). This study demonstrated that diastolic dysfunction is a significant predictor of adverse cardiac function independent of LVEF [13]. Another study by our group has previously shown that, in patients undergoing coronary artery bypass (CAB) and/or Aortic Valvular surgery, diastolic function evaluated by preoperative Transthoracic Echocardiography (TTE) predicts hospital stay in patients with varying degrees of systolic function [14]. That study did not address the predictive capacity of transesophageal echocardiography (TEE), in which parameters of left ventricular (LV) nonsystolic function (e.g., LA max volume) are much less reliably [15,16,17] or less practically [18,19] obtained. A recent study to evaluate the impact of CAD in diastolic function by echocardiography showed that the most common diagnostic marker of diastolic dysfunction with CAD is low e′ (e′—spectral pulse wave doppler at mitral annulus; lateral e′—68%; septal e′—53% patients) and LA index (>34 mL/m^2^), a marker of diastolic dysfunction recommended by American society of Echocardiography was elevated in only 18% of the study patients [20,21].

Many patients for CAB in our institution were referred from other peripheral hospitals with TTE or cardiac catheterization reports indicating significant multivessel CAD without actual images of TTE or catheterization. In those that did, very few had complete evaluations of nonsystolic function. However, all these patients would receive a TEE intraoperatively during CAB as standard protocol unless there is active contraindication. This prompted us to conduct a prospective cohort study involving subjects with normal systolic function undergoing CAB surgery to evaluate the association between intraoperative TEE metrics of LV performance during diastole and postoperative outcomes. The TEE algorithm depends on a limited number of metrics of performance during diastole and is validated to predict 5-year composite outcomes following CAB [22]. We hypothesized that the algorithm would be associated with duration of hospitalization after CAB.

## 2. Materials and Methods

### 2.1. Study Participants

Patients who underwent CAB using cardiopulmonary bypass (CPB) at the Johns Hopkins Hospital (JHH) from September 2017 to June 2018 and underwent a TEE examination to assess diastolic function were eligible for inclusion. Individuals were excluded for any of the following: left ventricular ejection fraction <50%; preoperative electrical pacing; inotropic support; or mechanical ventricular support. Institutional approval by the Johns Hopkins Medical Institutional Review Board was received, and requirement for written informed consent was waived. All study patients underwent CAB using CPB with cardioplegic arrest. None of the patients in the study had an off pump or beating heart surgery.

### 2.2. Echocardiographic Assessment

A protocol for evaluating performance during diastole by intraoperative TEE was introduced at JHH in January 2017 and intended for all patients undergoing CAB surgery. Examinations were performed after induction of anesthesia and prior to coronary artery revascularization, i.e., prior to CPB, though timing in relation to sternotomy or phase of respiratory cycle were not standardized. All TEEs were performed by physicians certified by the National Board of Echocardiography (NBE) in advanced perioperative TEE or by cardiac anesthesia fellows under the direct supervision of a certified physician. Metrics of performance during diastole were interpreted by a single physician (JMD) blinded to each subject’s preoperative comorbidities and postoperative course. The TEE evaluation of performance during diastole was modified from that of Swaminathan et al. [22] and included selected parameters of diastolic dysfunction: (1) spectral pulsed wave Doppler velocity of transmitral early (E) inflow; (2) spectral tissue Doppler imaging of diastolic myocardial velocity at the lateral mitral annulus (e′). Unless atrial fibrillation was present, the most representative waves were chosen. In the case of atrial fibrillation, values were averaged over 6–7 beats. Abnormal performance during diastole, hereafter referred to as diastolic dysfunction, was defined dichotomously as e′ < 10 cm/s. If LV dysfunction during diastole existed unlike during systole when it was normal, severity was defined as: (1) Grade 1 if E/e′ ≤ 8.5; (2) Grade 2 if E/e′ 8.6–12.5; (3) Grade 3 if E/e′ ≥ 12.6. These compromise E/e′ cutoffs were chosen to accommodate nonintegral values not specifically categorized in the reference manuscript [22]. Abnormal E was defined dichotomously as E > 50 cm/s [21]. These selected parameters of diastolic dysfunction (e′, E and E/e′) were chosen to represent diastolic function assessment as this study was solely based on intraoperative TEE.

### 2.3. Baseline Covariates and Follow-Up

Baseline (prior to surgery) demographic characteristics, comorbidities, and laboratory data were extracted from the medical record. Medical care throughout the hospital course was at the direction of the clinical team, who recorded diagnoses (ICD-10 codes), laboratory tests, and medications as part of routine care. The clinical team was not blinded to results of the evaluation of performance during diastole. Follow-up included the intraoperative and postoperative care phases until the time of hospital discharge. Complications were determined from abstraction of data routinely recorded in the medical record. Intravenous fluids and red blood cells administered volumes were abstracted from the medical record from time of surgery through the 4th postoperative day.

### 2.4. Outcome Measures

The primary outcome, time to hospital discharge, was defined as the time from end of surgery to hospital discharge. Secondary outcomes were: interval until extubation (requiring greater than 6 h following end of surgery), interval until free of supplemental oxygen (days from end of surgery to breathing room air), composite cardio-respiratory morbidity (i.e., postoperative diagnosis of heart failure, respiratory insufficiency, respiratory failure, or pulmonary edema), new atrial fibrillation, postoperative hypotension, hypervolemia, acute kidney injury (AKI; defined as a change of serum creatinine of 1.5 times baseline creatinine as per KIDGO criteria) [23], stroke, infection (surgical site infection [SSI], pneumonia, or sepsis), and readmission to the ICU.

### 2.5. Statistical Analysis

Continuous variables were summarized using mean ± SD. Categorical variables were summarized using percentage. Variables were compared across subjects with normal and diastolic dysfunction using *t*-test, Kruskal–Wallis, and chi-square test. Distinct clusters (heterogenous groups) within a patient sample were evaluated using latent class analysis (LCA) to classify subjects into multiple comorbidity groups based on comorbidities present at admission. LCA uses the joint distribution of observed responses across all individuals on a set of items (i.e., types of comorbidities) to characterize an underlying categorical latent variable that subdivides the given population into a smaller number of groups using modal class assignment. LCAs were conducted using the 12 baseline variables representing comorbidities based on cardiovascular diseases, chronic lung diseases, obesity, and baseline ASA score. Since the number of clusters is unknown a priori, statistical comparisons of model fit, based primarily on the log likelihood value and Bayesian Information Criterion (BIC), were used to compare models with an increasing number of clusters. An LCA is particularly suitable because of its ability to specify unobserved (latent) subgroups of individuals [24].

Cox Proportional Hazard models, with the response variable being ‘time to discharge’ and the event variable being ‘discharge’, were used to evaluate the association between time to discharge and diastolic dysfunction with and without adjustment for confounding variables. Additional Cox models were used to determine the role of secondary outcomes in mediating the relationship between diastolic dysfunction and time to discharge. Survival analyses were performed using the R programming environment. All estimates and confidence intervals (CIs) were obtained using the “*coxph*” function.

## 3. Results

The mean number of grafts that were revascularized was 3.35 ± 1 (median—3; mode—3). TEE evaluation of function during diastole was inconsistently completed, but consistency increased over the course of the study, approaching 90% of eligible patients by the end of the study. Relative frequency of reasons for incomplete evaluation (e.g., poor image alignment, excessive nonechocardiographic/clinical demands) could not be determined. Data were captured on 176 eligible subjects (mean age 65.2 ± 9.2 years, 73% male, 76% white). Baseline characteristics of study participants with abnormal (*n* = 105) vs. normal (*n* = 71) baseline performance during diastole are shown in Table 1. Those with abnormal selected echocardiographic parameters during diastole: (1) were older; (2) were more likely to be female, and/or have a history of congestive heart failure, valvular disease, and/or renal dysfunction; (3) trended toward a higher prevalence of prior myocardial infarction and history of chronic lung disease; (4) less likely to have hypertension; (5) trended towards a shorter duration of CPB. Overall, baseline characteristics suggested a greater burden of comorbidity at baseline for those with diastolic dysfunction, supported by significantly higher baseline ASA class, a composite comorbidity score well recognized to be associated with postoperative morbidity and mortality [25,26,27]. Based on the log-likelihood test and BIC, LCA classified subjects into two groups (high and low severity) depending on baseline characteristics, and the burden of comorbid illness appeared to be greater in latent class 2 than 1 (Table 1). Subjects with diastolic dysfunction based on selected TEE parameters were more likely to be classified in latent class 2 than 1 (Table 1).

Median time to hospital discharge was significantly longer for subjects with abnormal vs. normal performance during diastole (abnormal: 9.1/IQR 6.6–13.5 days; normal: 6.5/IQR 5.3–9.7 days) (*p* < 0.001 by Kruskal–Wallis test). Probability of hospital discharge on any given postoperative day (hereafter referred to as “daily probability of discharge”) was significantly lower for those with diastolic dysfunction (Figure 1). There was a dose–response relationship between severity of diastolic dysfunction and daily probability of discharge—those having the most severely abnormal performance had the lowest daily probability of discharge (Figure 2).

Adjusting for age, sex (Table 2, **Model 1**), duration of CPB (Table 2, **Models 1, 2**), severity of illness latent class (Table 2, **Models 2, 3**) history of heart failure and myocardial infarction (Table 2, **Model 3**), the daily probability of discharge remained significantly lower for those with diastolic dysfunction.

A sensitivity analysis excluding five subjects with more than mild valvular disease (e.g., MAC, aortic disease, mitral disease) did not affect these results (RR 0.63/95% CI 0.45–0.90). Our propensity analysis successfully matched 42 subjects with and without selected parameters of diastolic dysfunction (e′, E, E/e′), and showed that daily probability of discharge was 37% lower (RR 0.63/95% CI 0.40–1.00) for those with diastolic dysfunction. Our LCA showed the daily probability of discharge: (1) was significantly lower for the group with both diastolic dysfunction and high severity of illness latent class (RR 0.39/95% CI 0.26–0.58; *p* < 0.001); (2) tended to be lower in the group with diastolic dysfunction and low severity of illness (RR 0.72/95% CI 0.50–1.05; *p* = 0.09); (3) had no pattern of relation for the two strata with normal performance during diastole (Figure 3). Finally, in additional models that included LV performance during diastole, E and the E/e′ ratio as covariates in the regression model (Table 2, **Model 3**), the association between abnormal e′ and discharge was unchanged (RR 0.61/95% CI 0.42–0.88). Abnormal E was also independently associated with discharge (RR 0.61/95% CI 0.39–0.94), but E/e′ was not (RR 0.85/95% CI 0.48–1.52).

There were no significant differences in intravenous fluid administration (6169 ± 2958 mL vs. 6229 ± 3542 mL, *p* = 0.99) or transfusion of packed red blood cells (229 ± 404 mL vs. 187 ± 377 mL; *p* = 0.48) between subjects with and without diastolic dysfunction. Secondary intermediate outcomes are shown in Table 3.

Subjects with diastolic dysfunction were more likely to be extubated >6 h after surgery, to require supplemental oxygen longer, and to be diagnosed with the composite outcome of heart failure, respiratory insufficiency/failure, or pulmonary edema. We found no significant differences by performance during diastole in the incidences of postoperative atrial fibrillation, AKI, stroke, or infection. A higher proportion of subjects with diastolic dysfunction were readmitted to the ICU; however, this difference was not statistically significant (Table 3).

The relationship of selected parameters of diastolic dysfunction and secondary intermediate outcomes to daily probability of discharge is shown in Table 4.

Similar to baseline diastolic dysfunction based on TEE assessment of e′, E and E/e′, each of the secondary outcomes was associated with a lower daily probability of discharge in unadjusted analysis. In multivariable cox regression, secondary outcomes had modest effect on the relationship between baseline diastolic dysfunction and daily probability of discharge, which remained 37% less likely for those with selected parameters of diastolic dysfunction (Table 4).

## 4. Discussion

This prospective observational study evaluated the association between nonsystolic TEE metrics of left ventricular performance, assessed during intraoperative TEE before CPB, and interval to hospital discharge after CAB. We found that abnormal left ventricular performance during diastole was: (1) associated with a significantly longer postoperative hospital stay; (2) associated with a greater baseline comorbidity burden. Still, multivariate analysis showed their 35–40% lower daily probability of discharge (compared to subjects with normal performance during diastole) was independent of baseline comorbidities, including heart failure and myocardial infarction; (3) dose-dependently related to daily probability of discharge; and (4) associated with numerous complications after surgery, particularly heart failure and respiratory insufficiency/failure. These complications did not fully account for the relationship between diastolic dysfunction and daily probability of discharge.

Compared to patients with normal performance during diastole, we observed a 31% longer mean hospital stay in patients with diastolic dysfunction based on intraoperative TEE assessment of LV performance during diastole. This does not appear to be an artifact driven by outliers, given the nature of the nonparametric Kruskal–Wallis evaluation performed. The unadjusted ratio for daily probability of discharge in a patient with selected parameters of diastolic dysfunction is 0.54. This suggests a 46% lower daily probability of discharge if a patient has any degree of diastolic dysfunction (vs. normal performance during diastole). We have previously demonstrated that, independent of systolic function, preoperative TTE-measured diastolic dysfunction is associated with prolonged hospitalization as part of a combined endpoint in patients undergoing CAB, aortic valve replacement, or combined CAB and aortic valve replacement [14]. Severe diastolic dysfunction, determined by transmitral (E/A ratio, E wave deceleration time) and pulmonary venous flow patterns obtained by pre-operative TTE, has been shown to predict the occurrence of low output states within 30 days of CAB surgery [28]. The present study demonstrates that, in patients undergoing CAB surgery alone who have preserved systolic function, the presence of any degree of diastolic dysfunction based on selected intraoperative TEE parameters is associated with prolonged post-CAB hospitalization as an isolated endpoint.

The graded association between abnormal nonsystolic TEE metrics and prolonged post-CAB hospital stay that we observed in this study mimics the graded association between diastolic dysfunction and event-free five-year survival reported following CAB surgery [22]. Both studies had comparable patient comorbidity patterns and utilized similar echocardiographic analysis of performance during diastole. In our study, the prolongation in hospital stay persisted even after adjusting for all baseline co-morbidities (including heart failure). This suggests that classic co-morbidity focused pharmacotherapeutic interventions will fail to modify hospital stay following CAB in patients with normal systolic function but diastolic dysfunction (based on selected TEE parameters of diastolic function) in a manner similar to their failure to modify long-term morbidity in patients with preserved systolic function, diastolic dysfunction and a history of heart failure [29,30,31,32,33].

The relevance of the E/e′ ratio as an indicator of left sided filling pressures [21,34,35,36,37] is one of great debate and beyond the scope of our study. We note that, regardless of the E/e′ ratio, the presence of either E > 50 cm/s or e′ < 10 cm/s is associated with a lower daily probability of discharge following CAB surgery. We found no interdependence of E > 50 cm/s and e′ < 10 cm/s with regard to daily probability of discharge, suggesting that these individual metrics reflect distinct phenomenon. An E > 50 cm/s indicates that left atrial pressure is elevated in relation to LV diastolic pressure, regardless of ease of LV relaxation. Similarly, a slow e′ suggests that early LV relaxation is impaired, regardless of LA:LV pressure relationship [21]. The existence of either one of these conditions (E > 50 cm/s or e′ < 10 cm/s), regardless of the value of the other parameter, is associated with a lower daily probability of discharge in our study. The E/e′ ratio does not improve upon these associations.

There are several limitations to our prospective cohort study. First, protocol adherence was inconsistent over the course of the study. Limited feedback with respect to obstacles to adherence prohibits a true understanding as to the breadth of the protocol’s utility. The small study population restricts subgroup size, preventing utilization of a more robust statistical analysis. Thus, while our LCA and the propensity score analyses indicate that co-morbidities do not account for the entire effect of baseline diastolic dysfunction on hospital stay, a large number of our propensity score subjects did not have good matches. The study population size may also obscure the identification of an interaction between diastolic dysfunction and a perioperative morbidity. Finally, the true value of E/e′ remains unclear. Although originally defined as a noninvasive approach to evaluate left heart filling pressures [38], this concept has been challenged [36]. In our study, E/e′ does not add to the predictive capacity of E > 50 cm/s or e′ < 10 cm/s with regard to presence or absence of lower daily probability of hospital discharge following CAB, but does show a graded association with the degree to which daily probability of hospital discharge is lowered. It has previously been shown to correlate with long-term risk of major cardiac adverse effects [22]. In that study, as in ours, the study population was patients undergoing CAB surgery with baseline LVEF ≥ 50%.

## 5. Conclusions

In conclusion, we found that, compared to normal intraoperative TEE metrics during diastole prior to CPB, abnormal nonsystolic echocardiographic parameters are associated with a 31% longer hospital stay following CAB surgery in patients with normal baseline left ventricular systolic function. In this case, diastolic dysfunction is indicated by the presence of selected parameters—either E > 50 cm/s or e′ < 10 cm/s. The daily probability of discharge is inversely related to the severity of performance abnormality during diastole in a dose-dependent manner. The prolongation in hospital stay could be accounted for neither by baseline comorbidities nor by post-op complications. Larger studies are needed to confirm the consistency of these results and to elucidate a modifiable cause of the prolonged hospital stay.

## Figures and Tables

**Figure 1 jcm-11-03980-f001:**
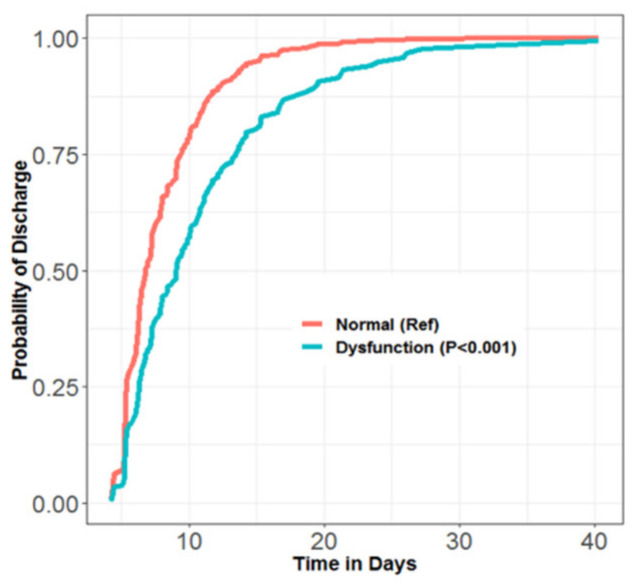
Cox proportional hazard model for daily probability of discharge vs. time (days) following surgery. Subjects with abnormal (**blue**) vs. normal (**red**) left ventricular performance during diastole.

**Figure 2 jcm-11-03980-f002:**
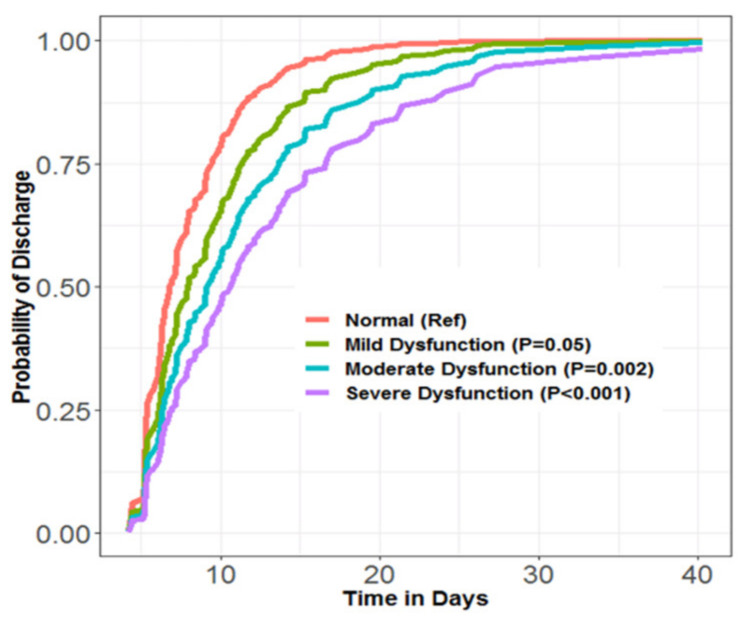
Dose–response relationship between severity of performance abnormality during diastole and daily probability of discharge; mild dysfunction (**green**), moderate dysfunction (**blue**), severe dysfunction (**purple**), normal (**red**). Survival analyses performed using R programming environment. Estimates and CIs obtained using the “*coxph*” function. Lower (RR 0.54/95% CI 0.40–0.75) for those with diastolic dysfunction compared to those with normal performance during diastole in unadjusted analysis.

**Figure 3 jcm-11-03980-f003:**
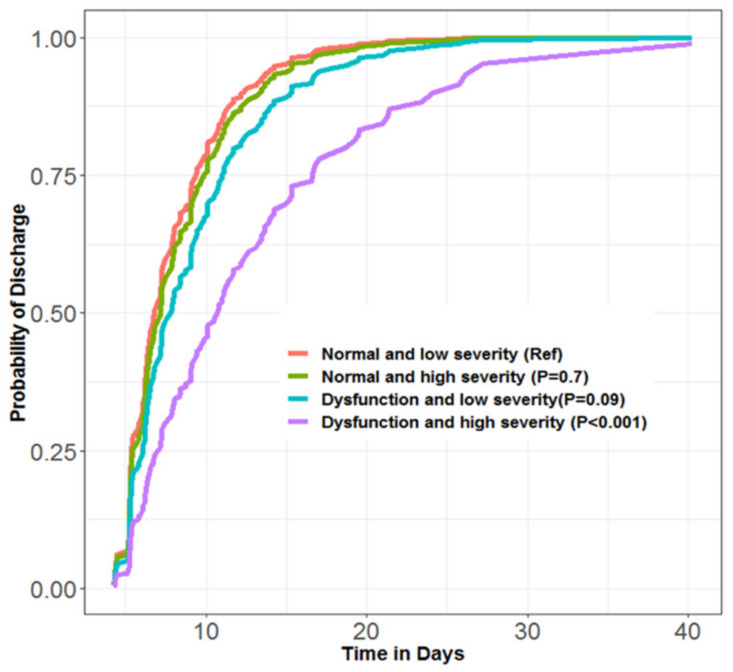
Probability of discharge over time in days by interaction between pre-surgical performance during diastole (normal/abnormal) and latent class (low/high severity) using the Cox proportional hazard model.

**Table 1 jcm-11-03980-t001:** Characteristics of study sample for subjects with abnormal and normal pre-surgical performance during diastole.

	Abnormal *n* = 105	Normal *n* = 71	*p*-Value
Age, yrs (SD)	67.5 ± 9.02	61.6 ± 8.5	<0.001 ***
Gender Female, *n* (%)	36 (34.2%)	11 (15.4%)	0.009 **
Race, N (%) White/Balck/Asian/others	80 (76%)/12 (11.5%)/ 9 (8.6%)/4 (3.8%)	54 (76%)/9 (12.8%)/ 4 (5.7%)/4 (5.7%)	0.86
Congestive Heart Failure, N (%)	21 (20%)	4 (5.6%)	0.01 *
Valvular Disease, N (%)	28 (26.6%)	8 (11.2%)	0.02 *
Peripheral vascular disease	15 (14.2%)	5 (7.0%)	0.21
Hypertension, N (%)	49 (46.6%)	52 (73.2%)	<0.001 ***
Diabetes mellitus, N (%)	40 (38%)	30 (42%)	0.69
Chronic lung disease	25 (23.8%)	8 (11%)	0.06
Chronic kidney disease	27 (25.7%)	7 (9.8%)	0.01 *
Obesity, N (%)	33 (31.4%)	22 (31%)	0.99
Myocardial infarction, N (%)	29 (27.6%)	10 (14.0%)	0.052 *
Stroke/TIA, N (%)	7 (6.6%)	4 (5.6%)	0.99
Arrhythmia, N (%)	26 (24.7%)	11 (15.4%)	0.16
ASA class = 4, N (%)	55 (52.3%)	23 (32.3%)	0.01 *
Duration of CPB CPB	96.33 ± 37.5	108.0 ± 49.09	0.09 *
Latent Class 2-High severity of illness	48 (45.7%)	12 (17%)	<0.001 ***

*—statistically significant; **—statistically very significant, ***—statistically extremely significant.

**Table 2 jcm-11-03980-t002:** Adjusted and un-adjusted rate ratios for discharge from the hospital using cox proportional hazard model.

	Un-Adjusted Rate Ratios	*p*-Value	Adjusted Rate Ratios Model 1	*p*-Value	Adjusted Rate Ratios Model 2	*p*-Value	Adjusted Rate Ratios Model 3	*p*-Value
Abnormal pre-surgical performance during diastole	0.54 *** (0.40,0.74)	<0.001	0.52 *** (0.36,0.75)	<0.0001	0.56 ** (0.39, 0.79)	0.001	0.61 ** (0.42, 0.88)	0.009
E; ref = normal	0.62 * (0.41, 0.94)	0.023	0.57 * (0.37, 0.88)	0.01	0.58 * (0.38, 0.88)	0.01	0.61 * (0.39, 0.94)	0.02
E-e′ ratio; ref = normal	0.57 * (0.33, 0.97)	0.04	0.78 (0.43, 1.42)	0.42	0.82 (0.46, 1.44)	0.49	0.85 (0.48, 1.52)	0.60
Duration of CPB	0.99 (0.99, 1.00)	0.61	0.997 (0.993, 1.00)	0.17	0.997 (0.993, 1.00)	0.12	0.996 * (0.992, 0.999)	0.044
Age	0.98 (0.97, 1.00)	0.08	0.993 (0.97, 1.01)	0.46				
Female gender	1.46 * (1.03,2.07)	0.03	1.05 (0.70,1.57)	0.80				
CHF	0.43 *** (0.26, 0.71)	0.001					0.50 * (0.25, 0.99)	0.047
Valvular disease	0.70 (0.47, 1.03)	0.07						
Hypertension	1.67 *** (1.22, 2.29)	0.001					1.14 (0.67, 1.92)	0.62
Chronic Lung disease	0.97 (0.66, 1.42)	0.88						
Renal disease	0.58 *** (0.40, 0.85)	0.006					0.55 * (0.30, 0.99)	0.049
Myocardial infarction	0.55 *** (0.38, 0.79)	0.001					0.51 *** (0.35, 0.75)	<0.001
ASA = 4 (ASA = 3 as Ref)	0.75 (0.55, 1.02)	0.06						
Latent class 2-high severity of illness	0.56 *** (0.40, 0.78)	<0.001			0.65 * (0.45, 0.93)	0.02	1.20 (0.56, 2.55)	0.63

*—statistically significant; **—statistically very significant, ***—statistically extremely significant.

**Table 3 jcm-11-03980-t003:** Secondary outcomes for subjects with abnormal and normal pre-surgical performance based on selected parameters during diastole.

	Abnormal *n* = 105	Normal *n* = 71	*p*-Value
Interval to extubation >6 h, *n* (%)	45 (42.8%)	17 (23.9%)	0.01 *
Interval to breathing from room air, days, median (IQR)	1.63 (0.68, 3.04)	0.89 (0.55, 2.82)	0.05 (Kruskal–Wallis)
Composite cardiopulmonary morbidity	21 (20%)	6 (8.4%)	0.037
Atrial fibrillation, N (%)	30 (28.5%)	14 (19.7%)	0.26
Hypotension, N (%)	47 (44%)	23 (32%)	0.13
Hypervolemia, N (%)	48 (46%)	34 (49%)	0.89
Maximum increase in creatinine, %, SD	18 ± 27	14 ± 15	0.21
KDIGO AKI, N (%)	26 (24.7%)	18 (25.3%)	0.99
Ischemic stroke, N (%)	0 (0%)	2 (2.8%)	0.16
Composite Infectious morbidity	6 (5.7%))	1 (1.4%)	0.23
ICU readmission, N (%)	8 (7.6%)	1 (1.4%)	0.08

Composite infectious morbidity = SSI, pneumonia, sepsis; Composite cardiopulmonary morbidity = heart failure, resp failure, resp insufficiency, pulm edema. *—statistically significant.

**Table 4 jcm-11-03980-t004:** Relation of baseline abnormal nonsystolic function and secondary outcomes to time to hospital discharge.

	Un-Adjusted Rate Ratios	*p*-Value	Adjusted Rate Ratios Model 1	*p*-Value
Dysfunction before surgery	0.54 *** (0.40, 0.75)	<0.001	0.63 * (0.45, 0.89)	0.01
Latent class 2-high severity of illness	0.53 *** (0.38, 0.74)	<0.001	0.73 (0.50, 1.07)	0.11
Duration of CPB	0.99 (0.99, 1.00)	0.70	0.997 (0.992, 1.001)	0.16
Interval to extubation >6	0.60 ** (0.44, 0.83)	0.002	0.81 (0.57, 1.15)	0.25
Time to room air	0.88 *** (0.81, 0.94)	<0.001	0.93 (0.86, 1.00)	0.08
Composite cardiopulmonary morbidity	0.47 *** (0.31, 0.73)	<0.001	0.78 (0.47, 1.30)	0.34
Atrial fibrillation	0.70 * (0.49, 0.99)	0.04	0.87 (0.60, 1.26)	0.49
Hypotension	0.61 ** (0.44, 0.83)	0.001	0.73 (0.52, 1.01)	0.06
Composite Infectious morbidity	0.30 ** (0.13, 0.70)	0.005	0.30 * (0.12, 0.77)	0.01
ICU Readmission	0.46 * (0.23, 0.90)	0.02	0.60 (0.29, 1.26)	0.18

Composite infectious morbidity = SSI, pneumonia, sepsis; Composite cardiopulmonary morbidity= heart failure, resp failure, resp insufficiency, pulm edema. *—statistically significant; **—statistically very significant, ***—statistically extremely significant.

## Data Availability

All data were de-identified and no personal data was stored. Data was stored in encrypted institutional computers and devices for analysis purpose. The datasets during and/or analyzed during the current study available from the corresponding author on reasonable request.

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
