# Peer review of "Selected Transesophageal Echocardiographic Parameters of Left Ventricular Diastolic Function Predict Length of Stay Following Coronary Artery Bypass Graft—A Prospective Observational Study"

_jcm, 2022, doi:10.3390/jcm11143980_

Round 1

Reviewer 1 Report

The authors did not contributed to most important issues of reviewers' comments. Moreover, they explained the study with the lack of TTE examinations in many patients performed at admission before surgery. 

I am sorry but the paper is not sufficient enough for the publication 

Author Response

Our inability to evaluate this is extremely disappointing to us as well. It would clearly add to the value of the manuscript. Unfortunately, we just do not have access to such data. Only about one third had an echocardiogram performed preoperatively (within 6 months). The vast majority of those did not address diastolic function in the report, and many of the preoperative TTE data we had was only a report (i.e., report from outside hospital without the actual TTE images in our system)Though a comparison with preoperative TTE would be nice, we do not feel the lack of this comparison invalidates our data.  

Reviewer 2 Report

Thank you for the corrections due to all comments.

The manuscript was really updated due to all comments .

Author Response

Dear Reviewer,

Thank you for your time to review our revised draft. We appreciate your comments and incorporating your suggestions have significantly improved the quality of the manuscript.  

Reviewer 3 Report

Mondal et al. present a prospective study in which they compared the number of postoperative days to discharge in patients undergoning CABG with and without diastolic dysfunction in intraoperative TEE. The study comprised a total of 176 patients wich were divided in two groups. They performed different models to asses and compare the influence of preoperative comorbidities in the number of days to hospital discharge. They found that abnormal non-systolic echocardiographic parameters were associated with a 31% longer hospital stay compared to normal non-systolic echocardiographic parameters, which was the primary outcome.

These findings seemed to be consistent when separating the patients into groups with low/high severity disease (preoperative comorbidities)  and normal/abnormal diastolic function.

When analyzing the secondary outcomes, patients with normal diastolic function had a shorter mechanical ventilation time, lower rate of composite cardiopulmonary morbidities and ICU re-admissions.

This manuscript was very well thought through, however I have some questions:

-The patients that were included in this study had a normal LV-systolic function, however there were patients with history of congestive heart failure. Are there any data describing the RV function?

-Did any of the patients have a history of STEMI or NSTEMI?

-When performing the surgeries, there is no description on the technique used, were all patients operated on with CPB or where there any patients who underwent surgery without CPB? For those who underwent surgery with CPB was a cardioplegic arrest used or where they operated on using a beating heart technique?

- What was the mean number of diseased vessels, number of bypasses performed and in which patients was a complete revascularisation achieved?

-Could you please describe, in the patients with valvular disease, what kind it is?

-In the limitations section you mention that at the beginning of the study there was difficulties with protocol adherence, what is meant by this? Who did not adhere to protocol, the physicians performing TEE, physicians including patients in the study? What impact did this have on these results?

-The fact that a propensity score match was performed to directly compare patients with and without diastolic dysfunction is, in my opinion a great idea, though its a shame, that only 42 subjects were matched.

-What do the sentences and paragraphs in red mean?

-In Table 3 its Kruskal-Wallis, not Wallace.

In my opinion there are many confounding variables that may lead to a prolonged lenght of stay at the hospital. Cardiac patients are usually multi-morbid and it is difficult to identify the one variable (like finding a needle in a haystack) responsible. This manuscript shows the importance that every step of the cardiac cycle may play an important role in patients with coronary heart disease. 

Author Response

Mondal et al. present a prospective study in which they compared the number of postoperative days to discharge in patients undergoning CABG with and without diastolic dysfunction in intraoperative TEE. The study comprised a total of 176 patients wich were divided in two groups. They performed different models to asses and compare the influence of preoperative comorbidities in the number of days to hospital discharge. They found that abnormal non-systolic echocardiographic parameters were associated with a 31% longer hospital stay compared to normal non-systolic echocardiographic parameters, which was the primary outcome. 

These findings seemed to be consistent when separating the patients into groups with low/high severity disease (preoperative comorbidities)  and normal/abnormal diastolic function. 

When analyzing the secondary outcomes, patients with normal diastolic function had a shorter mechanical ventilation time, lower rate of composite cardiopulmonary morbidities and ICU re-admissions. 

This manuscript was very well thought through, however I have some questions: 

-The patients that were included in this study had a normal LV-systolic function, however there were patients with history of congestive heart failure. Are there any data describing the RV function? 

Reply - This is an excellent question, and we have re-explored each patient’s chart individually to try to answer it. It is important to remember that, although some patients had a history of congestive heart failure, left sided systolic function was preserved or recovered at the time of the surgery (per our exclusion criteria of LVEF < 50%). It appears, from our review of right sided function as determined by the intra-operative anesthesia team (including a board-certified echocardiographer), only one patient had tricuspid pathology of moderate or greater degree. That one patient had moderate TR with good RV function patient and ended up with a prolonged post-operative course. None of the patients had RV dysfunction of moderate or greater degree. That does reinforce that our study included patients with normal baseline systolic function undergoing coronary artery bypass surgery. 

As a reminder, our exclusion criteria of the study were – 

  • Left ventricular ejection fraction <50% 
  • Preoperative electrical pacing 
  • Inotropic support 
  • Mechanical ventricular support. 

Did any of the patients have a history of STEMI or NSTEMI? 

Reply - This is another great question defining the population included in our study. We have included this in the baseline characteristics of the patients - 29 (27.6%) patients in abnormal diastolic function group and 10 (14%) patients in normal diastolic function group had myocardial infarction. (See Table 1) 

-When performing the surgeries, there is no description on the technique used, were all patients operated on with CPB or where there any patients who underwent surgery without CPB? For those who underwent surgery with CPB was a cardioplegic arrest used or where they operated on using a beating heart technique? 

Reply - Thank you for the question – we appreciate it. 

All patients underwent coronary bypass graft surgery on CPB with cardioplegic arrest. None of the patients in the study had an off pump or beating heart surgery. We have added this statement to the manuscript. 

What was the mean number of diseased vessels, number of bypasses performed and in which patients was a complete revascularisation achieved? 

Reply - This inciteful question, determining whether incomplete revascularization explains differences in post-operative hospital length of stay, can probably not be completely answered by the information we have. It was only very rarely that the surgeon made a comment that they knowingly left territories unrevascularized. It is also unlikely that all surgeons would agree as to which territories warrant the additional time on bypass to try to revascularize. In any case, we did determine the mean number of bypasses performed – it was 3.35 (SD 1); since there were some outliers to affect the mean, we also calculated the median and mode of number of grafts that study individuals received, and they were both 3. 

-Could you please describe, in the patients with valvular disease, what kind it is? 

Reply - We appreciate reviewer’s question. Valvular disease was mild in all but 5 cases. Three patients had moderate MAC, 1 had moderate aortic stenosis and one had moderate mitral regurgitation. When we exclude these 5 subjects in a sensitivity analysis, we again find small changes that do not alter our conclusion. This is conveyed in the manuscript. All study patients’ surgery was limited to isolated CABG.  

-In the limitations section you mention that at the beginning of the study there was difficulties with protocol adherence, what is meant by this? Who did not adhere to protocol, the physicians performing TEE, physicians including patients in the study? What impact did this have on these results? 

Reply - This is a great question – we appreciate the opportunity to explain ourselves.  

A standardized TEE protocol to assess diastolic dysfunction was introduced by the study PI. Adherence to the protocol by individual attending anesthesiologists, all of whom were NBE certified echocardiographers, grew gradually. Initially, enthusiasm for change was not uniform. As this inertia was overcome, standardized image quality was created by repeated input from the PI. We used doppler mediated diastolic function assessment which heavily relies on doppler shift angle and poor alignment can give a very inaccurate result. After the first four to six weeks of the study, protocol adoption approached 90%. Limited feedback with respect to obstacles to adherence prohibits a true understanding as to the breadth of the protocol’s utility. 

-The fact that a propensity score match was performed to directly compare patients with and without diastolic dysfunction is, in my opinion a great idea, though its a shame, that only 42 subjects were matched. 

Reply - We appreciate reviewer’s comment and recognize that our study sample was small. We have mentioned in the limitation and conclusion sections how our sample size has limited us many ways conducting subgroup analysis and having a larger sample size could add statistically significant impact of diastolic dysfunction on other perioperative morbidities. 

-What do the sentences and paragraphs in red mean? 

Reply - Thank you for asking – those were highlighted to respond to other reviewers’ comments. 

-In Table 3 its Kruskal-Wallis, not Wallace. 

Reply - Thank you for your thoroughness – we have corrected our mistake in the manuscript. 

In my opinion there are many confounding variables that may lead to a prolonged lenght of stay at the hospital. Cardiac patients are usually multi-morbid and it is difficult to identify the one variable (like finding a needle in a haystack) responsible. This manuscript shows the importance that every step of the cardiac cycle may play an important role in patients with coronary heart disease.  

Submission Date 

24 May 2022 

Date of this review 

21 Jun 2022 06:53:31 

Reply - Thank you for your comment – we highly appreciate it. 

Round 2

Reviewer 1 Report

The diastolic dysfunction confirming HFpEF cannot be evaluated only based on E and e'. If the authors insist on the study methodology please consider changing the definition to elected parameters of diastolic dysfunction. 

I would not define diastolic dysfunction all patients defined as so in the study based only on E and e'. 

also, please consider updating of the bibliography, there is lack of important citations including most recent HF guidelines

Author Response

Thank you for reevaluating our manuscript. We agree that TEE guided E and e’ are not the sole parameters to diagnose diastolic dysfunction. We have revised our manuscript and added ‘selective parameters of diastolic dysfunction’ in the methodology as well as in the rest of the manuscript as appropriate.

We have also updated bibliography incorporating most recent HF guideline. (Ref 8)

Reviewer 3 Report

The authors have adressed all the question sand comments in a remarkable manner.

Author Response

(The authors gave the same response as above.)
